# A qualitative evaluation of treatment fidelity alongside a pilot trial of a novel therapy for pediatric Inflammatory Bowel Disease

**Jenny L. Olson**[1]*, **Gisell Castillo**[1], **Amelia Palumbo**[1], **Megan Harrison**[2,3], **Ruth Singleton**[3], **Manoj M. Lalu**[1,4,5,6], **Dean A. Fergusson**[1,6,7], **Alain Stintzi**[8], **David R. Mack**[2,3,9], **Justin Presseau**[1,6,10]

1 Methodological and Implementation Research Program, Ottawa Hospital Research Institute, Ottawa, Ontario, Canada, 2 Department of Pediatrics, University of Ottawa, Ottawa, Ontario, Canada, 3 Children's Hospital of Eastern Ontario Research Institute, Ottawa, Ontario, Canada, 4 Department of Anesthesiology and Pain Medicine, The Ottawa Hospital, Ottawa, Ontario, Canada, 5 Department of Cellular and Molecular Medicine, University of Ottawa, Ottawa, Ontario, Canada, 6 School of Epidemiology and Public Health, University of Ottawa, Ottawa, Ontario, Canada, 7 Department of Medicine, University of Ottawa, Ottawa, Ontario, Canada, 8 Department of Biochemistry, Microbiology and Immunology, University of Ottawa, Ottawa, Ontario, Canada, 9 Children's Hospital of Eastern Ontario Inflammatory Bowel Disease Centre, Ottawa, Ontario, Canada, 10 Department of Psychology, University of Ottawa, Ottawa, Ontario, Canada

* jolson@ohri.ca

**Data Availability Statement:** Data cannot be shared publicly because, consistent with the informed consent process approved by the

## Abstract

### Background

Process evaluations conducted alongside clinical trials can improve understanding of treatment fidelity and provide contextual knowledge to aide interpretations of trial outcomes. We adopted a multiple-goals perspective to investigate treatment fidelity in two related pilot clinical trials of an adjuvant treatment for pediatric-onset Inflammatory Bowel Disease. This included a focus on barriers and enablers of performing trial-specific activities and of integrating those activities into daily life.

### Methods

We conducted one-time semi-structured interviews with a sub-sample of participants of the Resistant Starch in Pediatric Inflammatory Bowel Disease (NCT04522271) and Optimized Resistant Starch in Inflammatory Bowel Disease pilot trials (NCT04520594) and their caregivers (N = 42). The trials examined the effects of personalized food-derived resistant starches as an adjuvant therapy on intestinal microbiome functioning. Interviews were conducted within 3-months of participants completing or withdrawing from the trials. Interview guides with age-appropriate language were developed and pilot tested. Codes were identified inductively though conventional content analysis and then mapped to personal projects analysis, to explore how participants navigated between activities.

### Results

Three themes were identified. The first described the potential impact of living with inflammatory bowel disease and taking prescribed medications. The second described

Children's Hospital of Eastern Ontario Research Ethics Board (contact via REB@cheo.on.ca), participants were informed that their de-identified data would only be made available upon request. Data are available from the upon request to the Ottawa Methods Centre at the Ottawa Hospital Research Institute (contact via methodscentre@toh.ca, +1 613-761-4395). The contact details provided are a centralized institutional point of contact.

**Funding:** This work is supported by funding from the Government of Canada through Genome Canada https://genomecanada.ca/ and the Ontario Genomics Institute https://www.ontariogenomics.ca/ (OGI-149), the Ontario Ministry of Economic Development and Innovation https://www.ontario.ca/page/ministry-economic-development-job-creation-trade (Project 13440), and the Weston Foundation https://westonfoundation.ca/. (AS, DRM) DRM also received funding in part through Distinguished Clinical Research Chair Award from University of Ottawa https://www.uottawa.ca/en. The funders were not involved with the study design, data collection, analysis, decision to publish, or preparation of the manuscript.

**Competing interests:** AS and DM are co-founders of MedBiome, a clinical microbiomics company. The other authors have no competing interests to declare. This does not alter our adherence to PLOS ONE policies on sharing data and materials.

characteristics of trial-specific activities that might impact on their enactment, including perceived difficulty, and challenges following procedures or using trial materials. The third described the integration of trial-specific activities with school, work, household demands, and social, and extracurricular activities.

## Conclusions

Adjusting to living with inflammatory bowel disease and managing its treatment can impact trial participation. Integrating trial-related activities into daily life can be challenging, which could heighten perceptions of goal conflict. Findings can inform interpretations of trial outcomes and development of strategies for trial optimization and implementation of the adjuvant therapy into clinical practice.

## Introduction

Confidence in interpretations of clinical trial findings can be improved through an understanding of treatment fidelity; that is, the degree to which treatments were delivered, received, and enacted as planned [1]. When treatment fidelity is unknown it can be difficult to determine whether trial outcomes can be attributed to treatment effectiveness, or to confounding variables or weaknesses in trial design and/or implementation. Replication may also be impeded [1, 2]. Process evaluations are a method of assessing fidelity in complex interventions, and can facilitate identification of contextual factors that might impact on intervention effects and/or implementation [1–3]. Findings can inform the optimization of study design and evaluation when interventions are scaled, and the development of treatment-fidelity strategies for translation into clinical practice. Given the integral nature of treatment fidelity to behavioural intervention studies [1], we conducted a process evaluation of treatment fidelity alongside two related pilot clinical trials of an adjuvant therapy in pediatric Inflammatory Bowel Disease (IBD).

People living with IBD face diminished quality of life, through financial burden, physical impairment, and psychological distress [4]. The incidence and prevalence of IBD is increasing worldwide [5]. Prevalence is compounding (i.e., continually increasing due to stable incidence and low mortality rates) in the Western World, while incidence is increasing other 'newly Westernized' geographic regions where IBD has not been historically reported. [5, 6]. At the same time, the economic burden of the disease is also increasing [7]. Therefore, it is important to find treatments to improve outcomes in this population.

Emerging evidence from pre-clinical and clinical trials suggests food-based therapies may be beneficial in the treatment of IBD through their effects on the intestinal microbiome [8]. Whilst early evidence on the effects of resistant starches on IBD is promising, clinical treatment effects vary considerably across studies, and recent studies indicate variability in responsiveness to resistant starches between individuals. This suggests it may be beneficial to personalise resistant starch therapy to individuals, by matching types of resistant starches to host microbiota [8]. Therefore, further clinical trials are indicated.

Participants of clinical trials are often asked to perform trial-specific activities in the context of everyday life, and thus need to integrate those activities with those undertaken in the pursuit of non-trial related life goals (e.g., school, work, family, and social). Managing multiple goals requires ongoing prioritization and navigation between goals to facilitate goal attainment [9].

Goal pursuit is, therefore, competitive, with multiple goals competing for limited resources (e.g., time, energy). Understanding how trial participants prioritise and navigate between trial-specific activities and daily life activities can broaden understanding of the factors likely to affect intervention fidelity, beyond those more closely related to trial activities. Such knowledge can provide more comprehensive insight into how to support participants to manage clinical trial activities in the context of daily life.

### Study aim

The aim of this study was to identify barriers and enablers of the trial-specific activities experienced by children, youth, and their caregivers while taking part in two pilot clinical trials investigating the effects of personalised resistant starches on clinical outcomes in children and youths with IBD. Specifically, we aimed to identify barriers and enablers of performing trial-specific activities and those arising through the incorporation of these behaviours into daily life. Using personal projects analysis [10, 11] as a guiding conceptual framework, this information will provide insight into a range of factors that may influence treatment fidelity. These insights will provide important context for interpretations of clinical trial outcomes, and may inform optimization of future trial iterations, and ultimately, the integration of the adjuvant treatment for IBD into clinical practice.

## Materials and methods

### Setting and participants

One-time, semi-structured interviews were conducted with a sub-sample of participants of the Resistant Starch in Pediatric IBD pilot trial (RSP: NCT04522271) and OptiMized REsistaNt Starch in Inflammatory Bowel Disease (MEND: NCT04520594). Detailed trial information on both trials can be found on the clinical trials registry (https://clinicaltrials.gov RSP Identifier: NCT0452227; MEND Identifier: NCT04520594). Both trials aimed to determine if a plant-based personally optimized resistant starch would target the underlying cause of IBD and improve intestinal microbiome functioning in pediatric participants living with IBD. Children and adolescents that were newly diagnosed with IBD, had initiated an induction therapy, and demonstrated a clinical response by 2–4 weeks were eligible to take part in the RSP trial, whereas those previously diagnosed with IBD and clinically stable in the maintenance phase of their IBD treatment were eligible to participate in the MEND trial. Procedures for both trials were otherwise identical. The one-year pilot trials were based at an urban pediatric teaching hospital in Canada. Participants were randomized to an intervention or control condition for 6-months and followed for an additional 6-months. Participants, caregivers, and treating medical personnel were blinded to group allocation.

In addition to taking a powder (15g/day-30g/day) that was either a personalized resistant starch (i.e., treatment condition) or a placebo (i.e., control condition), trial activities also included: (a) keeping a daily diary of symptoms, powder intake, and any adverse events; and (b) collecting up to ten stool samples over the 6-month treatment period. Participants were also asked to continuing taking standard care medications as prescribed by their treating physician. Accurate interpretation of treatment effects and associated clinical outcomes of both trials depended on participants consistently enacting the three trial activities and adhering to standard care medications (collectively referred to as the 'trial-specific activities' hereafter). Therefore, an examination of the barriers and enablers experienced by participants and their caregivers while undertaking these activities provides an opportunity to build insight on anything that may have influenced treatment fidelity.

A comprehensive description of the methods of the present study can be found in the study protocol [12]. It was originally planned that the present study would include only participants and caregivers of the RSP trial. However, the more restrictive inclusion criteria of the RSP trial resulted in a slower recruitment relative to the MEND trial, which had less restrictive inclusion criteria and was conceived after the RS Study. For expediency, the research team extended recruitment in the present study to include participants and caregivers involved in the MEND trial.

Purposive sampling was used to recruit children (aged 8–12 years) and youths (aged 13–17 years) who had participated in the RSP and MEND trials, and their caregivers. We anticipated a heterogeneous sample, made up of children, youths, and caregivers, disease subtypes, time since diagnosis, symptoms experienced, and socio-economic characteristics. Consistent with guidance on sample size for interview studies, an upper and lower range of participants (15–30 interviews with a mix of children, youths, and caregivers) was determined based on the anticipated heterogeneity of participants, as well as the nature of the research question, study design, planned analysis, and available resources [13]. The exact number of participants within the predetermined range was decided by the research team during data collection, based on the breadth, depth, and richness of the data question, to ensure the research question could be answered [14].

Participation was voluntary, and participants and their caregiver(s) were eligible if they had the opportunity to take the resistant starches treatment or placebo for at least one month. Participants were recruited within 3-months of completing the intervention or withdrawing from the trials (between February 11, 2021 and January 24th, 2022). All participants who completed or withdrew from either trial were invited to participate, until the sample size was determined to be sufficient, and recruitment was ceased. Four participants of the RSP trial were not eligible to participate (did not take treatment or placebo for at least one month). One participant from the MEND trial declined to participate. No participant from the RSP trial declined to take part in an interview. Participants were randomly approached as they were seen/contacted by the clinical staff to ascertain permission for the research team to approach them. The CHEO research team met/contacted participants to explain the study and put interested participants, who agreed to have their contact information passed on, in contact with our colleagues at OHRI who were conducting the interviews for the Fidelity Study. Thus, those expressing interest were referred to an independent researcher (GC), who was not involved with the pilot trials and had no prior relationship with participants. At this stage, participants were provided with detailed study information, including an explicit description of the study aims.

Approval was obtained from the Children's Hospital of Eastern Ontario Research Ethics Board (ID # 20/65X), prior to study commencement, including approval of procedures for obtaining and documenting verbal consent or assent. Consent/assent forms were emailed to prospective participants prior to them meeting with the Research Coordinator (GC), who obtained verbal consent/assent from participants via video conference or telephone and documented the process on verbal consent forms. Verbal consent was obtained from all participants with the capacity to consent. Caregivers provided consent on behalf of children not yet able to consent, and assent was obtained from children/youths with capacity to do so. Where participants were not capable of providing assent, consent was obtained from caregivers, and only the caregiver participated in the interview. Trial participants aged between 7–11 years could elect to participate with their caregiver or separately (i.e., the caregiver and child each having a separate 30-minute interview). Youths providing consent could elect to participate in a sole interview or to include their caregivers in the interview.

Participants received a $25 gift card as compensation for participation, and parking costs were reimbursed when applicable. Study findings are reported in accordance with the

consolidated criteria for reporting qualitative research (COREQ) checklist [15]. Pseudonyms were used for all participants.

## Data collection and analysis

Participants completed an online survey of demographic characteristics and participated in interviews in-person, over the phone, or by video conferencing. Interview guides with age-appropriate language and content were developed and piloted with children, youths, and caregivers (see S1 File). The guides included open-ended questions and probes to elicit descriptions of barriers and enablers of the activities of interest. When requested, participants were provided a copy of the interview guide prior to participation.

Interviews were conducted between February 24, 2021 and January 19, 2022 (during the COVID-19 pandemic). Interviews were facilitated by a female clinical research coordinator (GC) with a Master of Arts in social psychology and qualitative methods, under the guidance of senior co-authors with expertise in health psychology, adolescent health, and pediatric IBD (JP, MH, DM). The interviewer had no prior relationship with participants. Interviews were audio-recorded and transcribed verbatim by a professional transcription service. To minimize participant burden, transcripts were not returned to participants for review.

Data were analysed using NVivo version 14 (QSR International Pty Ltd). The analysis was performed by a female analyst (JLO) with a PhD in Health Psychology, and experience in behavioural science and qualitative research methods. Initial codes were generated inductively via conventional content analysis, allowing for identification of novel perspectives [16]. Codes were then mapped to the 17 personal projects units of analysis (PPA) [10, 11]. Personal projects are sets of interrelated actions performed over time, to facilitate the pursuit of goals that are personally meaningful and salient to the individual. PPA can be applied to systematically evaluate how goals facilitate, interfere with, and are prioritized in relation to one-another. PPA offers a conceptual framework to understand the interplay that occurs when individuals pursue multiple goals. Themes were then developed by reviewing topics within each unit of analysis and grouping related topics into categories representing types of barriers and enablers of the trial-specific activities. Broad themes were conceptualized as overarching constructs (categories of barriers/enablers), and subthemes were conceptualized as groups of related, but distinct barriers/enablers within each overarching construct. Exemplar quotes best representing each sub-theme were then identified.

Analyses were verified by a second analyst (AP, expertise in health psychology, qualitative methods) to establish credibility, by reviewing preliminary themes and discussing differences in interpretations with the first analyst. Differences in interpretations were resolved through mutual agreement.

## Results

Thirty-five invitations to participate were sent to clinical trial participants and their caregivers. No response was received to six of the invitations. One interview was planned and later cancelled by the caregiver (no reason provided). Twenty-eight interviews were conducted, involving 42 participants. Sample characteristics are presented in Table 1. Participants included 15 youths ($M_{age}$ = 15.1 years; 60% male), seven children ($M_{age}$ = 9.6 years; 71% male), and 20 caregivers ($M_{age}$ = 45.3 years; 10% male). Caregivers were in attendance during all interviews with children, and nine interviews with youths. The average duration of interviews was 40 minutes (range = 13–77 minutes).

Three themes were identified: (Theme 1) Barriers and enablers related to living with IBD and managing treatment; (Theme 2) Barriers and enablers of performing the trial-specific

**Table 1. Sample characteristics.**

|  | Children (n = 7) | Youths (n = 15) | Caregivers (n = 20) |
|---|---|---|---|
| Age in years—Mean (range) | 9.6 (7–12) | 15.1 (13–17) | 45.3 (40–63) |
| Gender* = male | 5 (71%) | 9 (60%) | 2 (10%) |
| Caregiver present in interview | 7 (100%) | 9 (60%) | - |
| Marital status of caregiver |  |  |  |
| Married/common law |  |  | 15 (75%) |
| Separated |  |  | 1 (5%) |
| Widowed |  |  | 1 (5%) |
| No response |  |  | 3 (15%) |
| Number of children |  |  |  |
| 1 |  |  | 2 (10%) |
| 2 |  |  | 11 (55%) |
| 3 |  |  | 3 (15%) |
| 4 |  |  | 2 (10%) |
| No response |  |  | 2 (10%) |
| Caregiver highest level of education |  |  |  |
| Finished college/grad school |  |  | 13 (65%) |
| Finished high school/some college |  |  | 4 (20%) |
| No response |  |  | 2 (10%) |
| Caregiver employment status |  |  |  |
| Working full time |  |  | 14 (70%) |
| Working part time |  |  | 2 (10%) |
| Other |  |  | 1 (5%) |
| Not working |  |  | 1 (5%) |
| No response |  |  | 2 (10%) |

*Participants provided a free-text response to the question 'what is your gender.'

activities; and (Theme 3) Barriers and enablers of integrating the trial-specific activities into daily life. Each theme included 2–3 subthemes (summarized in Table 2). A complete list of initial codes, and mapping of codes to PPA units of analysis is presented in the S1 Data.

### Theme 1: Barriers and enablers related to living with IBD and managing treatment

Participants experiences of living with IBD and taking prescribed, standard care medications impacted on the trial-specific activities.

**Living with IBD.**   Some participants felt very distressed after receiving the IBD diagnosis:

*The start of his sickness was a shock . . . it was very, very hard for me. I couldn't believe that . . . his problem . . . was very serious. [I] didn't want to accept it . . . I had a very hard time, I was very, very sad. (Rosie, caregiver)*

Trial participation could be particularly challenging as children, youths, and caregivers adjusted to living with IBD.

*. . . you've got all these questions about that medication because the medications are strong ones. And you're already sort of saying like, okay, if this doesn't work, then what, or what are*

**Table 2. Summary of themes and sub-themes.**

| Main themes | | Sub-themes | | Personal Projects Analysis Dimensions |
|---|---|---|---|---|
| 1 | Barriers and enablers related to living with IBD and managing treatment | Barriers and enablers related to: | | - |
| | | | living with IBD | Positive impact Negative impact |
| | | | taking prescribed standard-care medications | Positive impact Negative impact |
| 2 | Barriers and enablers of performing the trial-specific activities | Barriers and enablers related to: | | - |
| | | | perceived difficulty/ease of tasks | Difficulty |
| | | | use of trial procedures and materials | Challenge |
| 3 | Barriers and enablers of integrating the trial-specific activities into daily life | Barriers and enablers of integrating trial-specific activities and: | | - |
| | | | school and work-related activities | Positive impact Negative impact |
| | | | household and family demands | Positive impact Negative impact |
| | | | social, sporting, and extra-curricular activities | Negative impact |

*the side effects . . . now we're going to be in a research study where he's putting something else in his body. So that's where my questions came, okay, is he going to have symptoms and side effects because I'm thinking, let's not make things worse, there's going to be side effects from the medication that he's on, there's going to be other pieces that we have to deal with. And now I'm going to put him into a study that he's going to also have more side effects or symptoms . . . it does make it a little difficult for sure. (Emily, caregiver)*

IBD symptoms also made it challenging for some to perform the trial-specific activities.

*I think that was hard to take [the starches] with my flare because like I had so many things going on. Like I couldn't even think about taking the starch just because there's just so much going on. So, I think that's kind of what made it harder. (George, youth)*

*. . . he doesn't have bowel movements that frequently so part of it is we need to make sure we catch it when it does happen . . . I guess the liquid's only good for five days or something . . . So, we were kind of worried that we weren't going to be in that window . . . (Jack, caregiver)*

Several youths/children were withdrawn from the trial due disease flare.

*We finished the study, and we didn't do it, though, for a full 6-months. It was only like four because they stopped us at that stage. I don't know, he had his next scope, and it was decided that we were stopping . . . (Catherine, caregiver)*

On the other hand, when children and youths felt better, it was easier for them to take the resistant starches. When asked about taking starches, one youth said:

*It's pretty easy . . . every morning I took it and I'd always, I personally think that it helps me a lot. Ever since I got off of it, I feel a little different . . . Yeah, forgot it twice. Because, yeah, I forgot it twice. I just completely forgot. And then the days I forgot it, I didn't feel good. (Jake, youth)*

One child felt more in control of his illness when performing the trial-specific activities.

*I think [taking starches and completing the diary] gave him more control over the disease, being able to be in control of what was happening . . . So that was one thing that he could do to, to kind of help himself manage his systems, his symptoms, and experiences. (Marie, caregiver)*

**Taking prescribed standard-care medications.**   Taking standard care medications sometimes interfered with the other trial-specific activities. For example, medication side effects could make it challenging to take the resistant starches:

*So, when he takes methotrexate once a week, it can make him a little bit nauseous . . . there were a couple of times where he would have his methotrexate and feel nauseous afterwards, and then not want to have the smoothie [mixed with the resistant starches/placebo] . . . (Julia, caregiver)*

In some instances, medications were prioritized as more important than adherence to the adjuvant treatment:

*Giving her that resistant starch was by far the hardest thing. Just because . . . she didn't want anything to do with it . . . she's still trying to adjust to the meds, which in my mind, those were more important at this point to try and like control her Crohn's. So then just trying to get her to take another thing on top of that was just, yeah, too much. (Morgan, caregiver)*

Conversely, some participants found that pairing the starches with standard care medications, facilitated both activities.

*[Taking medications] definitely made it easier to remember [the starches] because. . . it just felt a lot easier to remember to take my medication. So, because I was taking them at the same time . . . taking the starch was just sort of like oh, I'll take this after my meds. It became a lot easier to remember to do. (Adam, youth)*

## Theme 2: Barriers and enablers of performing the trial-specific activities

**Perceived difficulty/ease of tasks.**   Perceived difficulty of the trial-specific activities presented a barrier for some participants, particularly early on. Many said the activities were difficult initially but became easier over time.

*For me in the beginning, everything was hard . . . I don't know what to think . . . It's very hard for me. Because the stool sample I do only one time a month. Not bad. And starch every day is like kind of the responsibility like, become easier, everything was become easy, now for me, I am almost in the end. It's everything easy for me. (Angela, caregiver)*

However, some did not find the trial-specific activities to be difficult at all, and therefore perceived difficulty was not a barrier for these individuals. For instance, when asked if anything could have made things easier, one caregiver said:

*I don't think so . . . it was pretty straightforward, there wasn't a whole lot. It was pretty straightforward. (Andrea, caregiver)*

**Use of trial procedures and materials.** Some participants faced challenges using trial materials or following trial procedures. For some, the amount of powder (i.e., starch or placebo) to be taken each day was challenging.

*. . . it was like a good quantity, and they said that you could break it up and put it into several different things, which we tried, but then that just prolonged our torture. Because now we're like trying to get it into her several times a day instead of just once. So, then you're like fighting with her constantly . . . it didn't work. (Morgan, caregiver)*

The duration of time that the adjuvant therapy was to be taken was also a barrier for some:

*Dwayne got a little bit sick of having a smoothie every single night for like six months . . . it does like maybe after month five get to be a bit of a grind to like, okay, one more month to go, it was like that, you know, seeing the finish line, but, like, needing that little extra to get through. (Julia, caregiver)*

Some participants described uncertainty around what to record in the diary. When asked if anything might make trial participation easier, one person said:

*. . . maybe a little bit more instructions on . . . how much information would be helpful to be collected. Like . . . how would the researchers like me to keep that journal? Because I felt like I wasn't doing as good of a job. (Jane, caregiver)*

Feeling uncomfortable using the equipment for stool sample collection was another challenge.

*I think just collecting [stool samples] to begin with, with the hat thingy, the white piece that you put under the toilet seat. I've never enjoyed using that like ever since I was young . . . It, for some reason just makes me uncomfortable to use it. . . . It kind of like distracts you, but everything else was fine. (Martina, youth)*

More generally, the overall burden of trial participation was a challenge for some.

*There was a lot that went into it in the background . . . making sure we never ran out of food, or making sure the blender was charged, and is your phone charged? And if we're going somewhere, do we have the starch with us, and do we have the stool sample, do we have the kits? . . . there was a lot of little background things that put a lot more stress on as the parent. (Ashley, caregiver)*

Conversely, having a flexible window of time in which to collect the stool sample was considered helpful by many participants.

*The stool sample collection always had a range of time. So, if the box came on a Wednesday, the instructions were to collect it on a Sunday or a Monday or a Tuesday. So, there was flexibility on when it could be collected . . . So, it was very convenient in that regard. (Francis, caregiver)*

## Theme 3: Barriers and enablers of integrating the trial-specific activities into daily life

When trial participation was highly prioritized, participants experienced lower levels of interference between trial-specific activities and other life activities.

*[Prioritizing trial participation] was definitely something I made sure I did. If something really important had come up, then maybe I wouldn't have been as consistent with it and I might've, you know, focused on something else. But there was nothing like that particularly interfered with doing this. So, you know, it stayed something that I made sure I did. (Adam, youth)*

However, most participants experienced some interference between the trial-specific activities and aspects of daily life. In general, being away from home for any reason presented a barrier to the trial-specific activities. When discussing missed starches, one participant said:

*. . . especially if you're in the middle of something and if you break the routine, like if we'd go to like grandparents during the summer and, or there's some things happening, then you just forget. (Jane, caregiver)*

However, pandemic-related stay-at-home advice often minimized the impact of this barrier, thereby enabling the trial-specific activities.

*. . .it would have been harder to do if there wasn't COVID because he had to be home all the time. It made it easy for me, therefore, to keep on him and with reminders and make it accessible. I would probably be answering very different if we'd had a normal life, because we are out, we leave on weekends and go to hockey tournaments and visit friends and I'm thinking it might've got a little harder to remember every day and bring stuff . . . And like for this study it was very helpful because he had to stay home. So, it made it very easy for him to get back into that routine and follow it and do the study. (Catherine, caregiver)*

**School and work-related activities.**   School activities sometimes interfered with the trial-specific activities.

*. . . I was dealing with my last year of high school and it's really frustrating because I don't have a lot of time to be making a smoothie. . . things just got really, really busy . . . it just became more like a hassle . . . it felt more like homework. Because before when I started, I wasn't as busy . . . But then all of a sudden, my teachers just start laying off a bunch of work . . . and that's kind of when things to start going south. (Blossom, youth)*

Balancing a combination of school, work, and the trial-specific activities could be particularly challenging for youths.

*When I work, sometimes I don't have the time (Cleopatra, youth).*

*She missed, in three or four months now I believe, she missed [the starches] three or four times, she started working recently so she gets back from school at 3.30, has to be at work for four, and comes back at nine. So, I believe that for a couple of days, she would have missed it because her schedule was too tight. (George, Cleopatra's caregiver)*

Work demands also made it more challenging for caregivers to support the trial-specific activities.

*I'm a nurse . . . So, it did make it hard . . . when I'm doing shift work. I'm not here at the same time every day, it's sort of all over the board. So, that's hard in many areas of life where you*

*need to do things routinely. So that was definitely a complication or something that was hard in our lives. (Sophie, caregiver)*

For one caregiver, managing multiple competing demands, including work and household activities influenced her willingness to take part in the trial:

*I think if I was at home, it would be easier . . . I was working evenings or like evenings, weekends, daytime hours, and it was all scattered. Plus having to pair all the food and meals to make sure that they were all ready for [child] . . . plus having a teenager finishing his last year at high school. It was, it was a lot. So, I think trying to juggle everything was, I just, I wasn't myself as a willing participant. I wasn't as organized, haven't cleaned the house in over a year, like I wasn't as organized as I would have liked to have been in the study (Marie, caregiver)*

On the other hand, school and work arrangements sometimes enabled enactment of the trial-specific activities. For instance, the opportunity to earn credits for volunteering by taking part in the trial motivated several youths to participate. When discussing what motivated her daughter to complete the trial-specific activities one caregiver said:

*. . . Martina's very studious and she's very eager to have these volunteer hours, which was a huge incentive for her . . . She's going to get these hours and that's pretty important for her resume when she starts applying for school. (Andrea, caregiver)*

**Household and family demands.** Household activities sometimes interfered with the trial-specific activities. One participant said that being in a bad mood made it difficult to do stool sample collection, and to take the standard care medications. When asked what affected her mood, she described the demands of having to care for her younger brother:

*I like to sleep a lot and I found if I didn't get enough sleep, I wasn't motivated to do anything . . . Just getting up early because I have a little brother that can't watch himself and my mom has to work right at eight o'clock. So, I had to be up at 7:30 and I couldn't fall asleep for until really late at night. So having that little bit of sleep didn't make me happy. (Rosie, youth)*

When asked if other priorities interfered with trial participation, one youth said:

*I guess it would be like family wise . . . after I lost my sister, I realized how important my family is. And nowadays, it's like gotten worse, like my great uncle is in the hospital on the verge of dying. And then my stepmom recently got breast cancer. So, family has been more important than most things. (Blossom, youth)*

Sometimes the trial-specific activities and household demands were mutually conducive.

*. . . [taking starches] was useful because then one less thing to worry about taking for lunch. . . I'm making my own lunches and I thought it would just be so much easier if I knew exactly at least part of what I was going to take every day. So, then I wouldn't have to worry about it the next morning. (Elizabeth, youth)*

Supporting the trial-specific activities was consistent with one mother's approach to parenting. When asked if she had advice for trial future participants, she said:

*. . . with everything that I do with my kids, because as a mom, you kind of say you're the chauffeur, you're the cook, you're the cleaner, you're the doctor, you're the counselor, all that stuff. . . . I would say a silly thing like think of it like building a sandcastle or no, I'm today I'm a researcher, like we would just try and make [stool sample collection] amusing, so that we could get it done . . . (Marie, caregiver)*

**Social, sporting, and extra-curricular activities.** Social, sporting, and extra-curricular activities sometimes interfered with the trial-specific activities.

*The time I kind of find difficult is if like I wanted to go to a friend's house. It's not really the first thing in my mind is to remember to take my starch before I go out. So not that often, but there was times I forgot to take it . . . and sometimes it can be annoying to go mix it up in something if you're like in a rush . . . (Keith, youth)*

For some, multiple demands from work, school, and extracurricular activities made it particularly challenging to perform the trial-specific activities.

*. . . the previous [stool] samples that Julia had to provide; she was still at home. She wasn't in school all day and she was working part-time hours or whatever. So, it was easier to fit into her schedule. This last one she has after school activities, plus she has a job, plus she's going to school full-time. So, I think that it just was a little bit of extra stress, but she wanted to make sure she can meet that deadline. (Andrea, caregiver)*

## Discussion

This study identified factors that may have impacted on treatment fidelity in the RSP and MEND pilot trials of an adjuvant resistant starches therapy for pediatric IBD. The work provides contextual information useful for interpreting trial findings. The findings can also inform optimization of study design and evaluation in future iterations of the clinical trials, and the development of treatment-fidelity strategies for translation of the adjuvant therapy into clinical practice. As with other chronic conditions, children with IBD may face a variety of challenges adhering to adjuvant therapy. For instance, children and adolescents diagnosed with type 1 diabetes mellitus are advised to follow dietary recommendations in addition to taking prescribed insulin [17]. However, adherence to dietary recommendations is low, with adherence levels between 21% and 95% across eating behaviours. Poorer adherence is associated with parent-child mealtime behaviours and poor knowledge of dietary management for diabetes.

We identified three major themes, representing potential threats to treatment fidelity. The first theme provides insight into the experiences of young people and caregivers as they learn to navigate life with IBD and manage its treatment. Adherence to the adjuvant therapy and other trial-specific activities may be difficult as children, youths, and caregivers face these challenges. The second theme encompasses more direct challenges and facilitators of performing trial activities. Clinical trial participation may be difficult for some people at first but is likely to become easier over time. On the other hand, intervention fatigue may be an issue for some individuals towards the end of the active treatment period, while participants may experience burden at any stage of the intervention. Strategies should be tailored to suit the evolving needs of intervention participants as they progress through the 6-month active treatment period. Minor amendments to trial procedures (e.g., more detailed instructions on what to record in the diary), and strategies to support participants to overcome negative evaluations of

intervention materials (e.g., amount of powder to be consumed) may also support adherence. The final theme provides insight into how trial participants navigate between trial-specific activities and activities of daily life and demonstrates the importance of considering a broader range of factors that may impact on treatment fidelity.

Consistent with previous research [18], adjusting to a diagnosis of pediatric-onset IBD was difficult and distressing for some participants in the present study. Adherence to adjuvant resistant starches and related clinical trial activities may be hampered as participants and caregivers come to grips with living with IBD, especially when young people feel unwell. Furthermore, side-effects from standard-care medications may present an additional barrier. This is perhaps unsurprising, given previous research indicating side-effects and feeling unwell can interfere with medication adherence in the treatment of pediatric-onset IBD [19]. We extend these findings to show that these experiences can also impact on adherence to adjuvant therapy. In the present study medications for initial treatment and ongoing maintenance of IBD were more problematic than the addition of the adjuvant therapy. It is important to be aware of these challenges and find strategies to support the integration of adjuvant therapy and clinical trial activities even when participants are feeling distressed, unwell, or experiencing side-effects from their prescribed medications.

Activities performed in the pursuit of school, work, household, social, sporting, and extra-curricular goals presented barriers to performing the trial-specific activities. Consistent with research into barriers of medication adherence among adolescents with IBD [20], being away from the family home was a common barrier to taking resistant starches. Clinical trials often necessitate the integration of trial activities into daily life. Aspects of individuals' daily lives may interfere with, or support treatment fidelity. However, evaluations of treatment fidelity typically focus on identifying more direct barriers and enablers of performing trial activities, without consideration of factors that may influence the integration of trial activities into daily life. Our findings reinforce the need to consider the enactment of health behaviours (e.g., taking resistant starches) alongside other behaviours performed in the pursuit of personally salient goals (e.g., doing homework to achieve good grades in school or visiting with friends to have a fulfilling social life) [11]. Such investigations can provide a more comprehensive overview of treatment fidelity, which is important for the interpretation of trial outcomes [1, 2].

Perceptions of goal conflict are negatively associated with goal attainment [9]. Therefore, clinical trial participants who experience higher levels of goal conflict may be less likely to achieve their goals, including those related to trial participation. We identified multiple instances of goal conflict in the present study. Trial-specific activities and daily life activities interfered with one another. Moreover, the demands of multiple daily life activities sometimes compounded to further impact on trial-specific activities. This represents a threat to treatment fidelity, warranting strategies to minimize goal conflict. Supporting clinical trial participants to prioritize and navigate between multiple competing demands could help them to minimize goal conflict. Coping planning (i.e., preparing to overcome anticipated barriers to action) is an effective strategy for supporting sustained health behaviour change, particularly when intervention participants are supported to make those plans [21, 22]. For instance, youths may be less likely to miss resistant starches doses if they are encouraged to plan when, where, and how they will take the starches on days when trial, school, and work commitments converge.

The application of a multiple goals perspective to support identification of a wider range of proximal and distal factors that may impact treatment fidelity in a clinical trial was a strength of this study. These methods could be utilized to conduct more comprehensive process evaluations alongside other clinical trials. The simultaneous conduct of a process evaluation alongside a pilot clinical trial is another strength of this study and will inform interpretations of future findings on treatment effectiveness. The findings can also inform intervention

replication, scale up, and implementation into clinical practice. The interviews of children and youths was another benefit of the study, providing unique views and perspectives about clinical trial participation.

Despite the strengths of this study, there are also limitations to be reported. The process evaluation was conducted during the COVID-19 pandemic, and participants indicated that pandemic-related changes in daily routines impacted on the enactment of trial activities. It is possible that barriers and enablers of trial-specific activities differ in post-pandemic life, and findings should be interpreted accordingly. Furthermore, goal systems are dynamic, changing as individuals navigate between competing goals and priorities [11]; however, the study design (i.e., interviews conducted at a single timepoint) precluded assessment of how participant's goals and priorities changed while taking part in the clinical trials. Given the 6-month active intervention period, it is also possible that participants were able to more easily recall their experiences toward the end of the intervention, compared to early on. Although many participants provided detailed descriptions of the challenges faced upon commencing the intervention, the possibility of recall bias cannot be ruled out. Future studies could assess barriers and enablers of trial activities across multiple time points to improve understanding of the effects of dynamic goals systems on treatment fidelity over time.

It is also possible that the mental health of participants may have influenced study findings. Children with chronic physical illnesses experience greater risk of psychiatric disorders compared to those without a chronic illness [23] and there is a higher incidence of mental illness, including depression and anxiety, in people with IBD compared to matched controls [24], especially in the first year of diagnosis [25]. For some participants, managing co-occurring conditions is a very important goal of daily life. Although we did not include specific questions to surface barriers related to co-occurring conditions, the interview guide was broad enough to facilitate discussion of any factor considered salient by participants.

## Conclusions

Application of a multiple goals perspective facilitated identification of a comprehensive range of factors likely to impact treatment fidelity. These included the experiences of individuals as they adjusted to living with and managing IBD, challenges performing the trial-specific activities, and challenges integrating trial-specific activities into daily life. Activities performed in the pursuit of health, school, work, household, social, sporting, and extra-curricular goals sometimes interfered with trial-specific activities. Interference between these activities could heighten perceptions of goal conflict, which may threaten treatment fidelity. The findings should be considered when interpreting effectiveness outcomes of the clinical trials and can inform the selection of strategies to support children and youths to overcome barriers and leverage enablers of taking resistant starches and enacting other trial-specific activities. This could optimize treatment fidelity in future trials (replication and scale-up), and when implementing the adjuvant therapy into clinical practice.

## Supporting information

**S1 File. Interview guides.**
(DOCX)

**S1 Data. Coding tree.**
(XLSX)

## Author Contributions

**Conceptualization:** Megan Harrison, Manoj M. Lalu, Dean A. Fergusson, Alain Stintzi, David R. Mack, Justin Presseau.

**Data curation:** Gisell Castillo, Megan Harrison.

**Formal analysis:** Jenny L. Olson, Amelia Palumbo.

**Funding acquisition:** Alain Stintzi, David R. Mack.

**Investigation:** Gisell Castillo, Megan Harrison, Manoj M. Lalu, Dean A. Fergusson, Alain Stintzi, David R. Mack, Justin Presseau.

**Methodology:** Gisell Castillo, Megan Harrison, Manoj M. Lalu, Dean A. Fergusson, Alain Stintzi, David R. Mack, Justin Presseau.

**Project administration:** Gisell Castillo, Ruth Singleton.

**Resources:** Alain Stintzi, David R. Mack, Justin Presseau.

**Supervision:** Justin Presseau.

**Writing – original draft:** Jenny L. Olson.

**Writing – review & editing:** Jenny L. Olson, Gisell Castillo, Amelia Palumbo, Megan Harrison, Ruth Singleton, Manoj M. Lalu, Dean A. Fergusson, Alain Stintzi, David R. Mack, Justin Presseau.

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
