## [Decision Letter · Decision Letter 0]

14 Nov 2023

PONE-D-23-30679A qualitative evaluation of treatment fidelity alongside a pilot trial of a novel therapy for pediatric IBDPLOS ONE

Dear Dr. Olson,

Thank you for submitting your manuscript to PLOS ONE. After careful consideration, we feel that it has merit but does not fully meet PLOS ONE’s publication criteria as it currently stands. Therefore, we invite you to submit a revised version of the manuscript that addresses the points raised during the review process.

We look forward to receiving your revised manuscript.

Kind regards,

Yasin Sahin

Academic Editor

PLOS ONE

Journal Requirements:

2. In the ethics statement in the Methods, you have specified that verbal consent was obtained. Please provide additional details regarding how this consent was documented and witnessed, and state whether this was approved by the IRB.

"AS and DM are co-founders of MedBiome, a clinical microbiomics company. The other authors have no competing interests to declare. "

Reviewers' comments:

Reviewer's Responses to Questions

**Comments to the Author**

1. Is the manuscript technically sound, and do the data support the conclusions?

Reviewer #1: Yes

Reviewer #2: Yes

2. Has the statistical analysis been performed appropriately and rigorously? 

Reviewer #1: Yes

Reviewer #2: Yes

3. Have the authors made all data underlying the findings in their manuscript fully available?

Reviewer #1: Yes

Reviewer #2: Yes

4. Is the manuscript presented in an intelligible fashion and written in standard English?

Reviewer #1: Yes

Reviewer #2: Yes

5. Review Comments to the Author

Reviewer #1: Jen L. Olson, et al. investigated treatment fidelity in two related pilot clinical trials of an adjuvant treatment for pediatric-onset Inflammatory Bowel Disease. This is an important topic and very relevant in research and clinical practice. Authors have done an excellent job conceptualizing framework and the study team had access to a niche group of patient participants and have conducted interviews to understand barriers and enablers of performing trial-specific activities and of integrating those activities into daily life. The manuscript is well-written.

I have some suggestions to consider

- Consider revising introduction section to focus on treatment fidelity and process evaluations, and its impact on IBD. IBD pilot trials and participants information can be included in the methods section.

- It is not very clear how did the authors identify main themes and subthemes. Authors may want to elaborate on this in study methods.

- Did authors study other factors that could have influenced the study findings, like mental health diseases including anxiety, depression, etc, and other comorbid medical conditions in the participants and caregivers?

- The discussion section needs a bit more work. Authors may want to include more evidence from other chronic conditions similar to IBD, and how patient experiences impact adherence to adjuvant therapy.

Reviewer #2: Review Comments to the Author

Thank you for inviting me to review this manuscript reporting the qualitative evaluation of treatment fidelity alongside two related pilot trials of targeted therapies for IBD. The paper reports interesting and original findings that can help inform the interpretation of future clinical trials, with regard to trial outcomes and improving methods, and methodologies.

The authors have conducted appropriate qualitative methodology, including thematic analysis, and particularly relevant is the adoption of the multiple-goals perspective to interpret the themes. I have no major concerns on the manuscript, however the authors do need to provide greater explanation of the relation between trials, justification for sample size, and acknowledgement of study limitations.

My specific comments are as follows:

Title

Spell out acronym ‘IBD’ in the title.

Abstract

There are scant details on the data collection methods. To the unversed, it is also unclear as to what ‘personal projects units of analysis’ refers to. You may wish to remove the ‘units of’, to provide standard PPA. Or succinctly explain/provide example.

Introduction

Page 4, lines 78-79: please explain briefly, why the prevalence of IDB is increasing worldwide.

Very good rationale and explanation of the context of the qualitative (nested) research within, the larger project (i.e., two related trials). You may, however, wish to give the reader insight into the respective aims of the two trials. Although you introduce both trials on page 4 (lines 83-84), you only state what they both examined.

Method

Page 7, lines 141-143: greater justification is needed to the sample size. Between 15-30 interviews is a broad range; does this also include carers/parents, or just trial participants (e.g., children)?

Lines 146-148: recruitment process – although you state when recruitment occurred, and that the research co-ordinator contacted potential participants, but there are no details as to how participants were selected, and from what database/recruitment pathway. These need describing.

Also consider including author initials for specific tasks in the Method; you only do this occasionally and inconsistently. For example, the final sentence (page 9, lines 189-190) does not explain who resolved differences in interpretation.

Results

Effective use of quotes to support the generated themes. These align to the study aims and map out in the Discussion.

Discussion

Page 22, lines 471-472: ‘personally salient goals’; this is ambiguous. Please provide brief example(s) from your original findings.

The authors provide no limitations to this qualitative, nested research. Please provide on revision.

6. PLOS authors have the option to publish the peer review history of their article (what does this mean?). If published, this will include your full peer review and any attached files.

Reviewer #1: No

Reviewer #2: **Yes: **James Peter Gavin

---

## [Author Response · Author response to Decision Letter 0]

3 May 2024

Journal Requirements:

RESPONSE: We have reviewed the publication’s style requirements, and confirm the manuscript is presented accordingly. 

2. In the ethics statement in the Methods, you have specified that verbal consent was obtained. Please provide additional details regarding how this consent was documented and witnessed, and state whether this was approved by the IRB.

RESPONSE: Participants in both the Resistant Starch Trial and MEND Study were offered the study up until recruitment was complete for this fidelity sub-study. Following CHEO RI guidelines for the consent process and approach, participants were randomly approached as they were seen/contacted by the clinical staff to ascertain permission for the research team to approach them. The CHEO research team met/contacted participants to explain the study and put interested participants, who agreed to have their contact information passed on, in contact with our colleagues at OHRI who were conducting the interviews for the Fidelity Study. Verbal consent was witnessed and documented on a verbal consent form by the research coordinator. The consent/assent processes were approved by the CHEO Research Ethics Board. This information has been added to the Methods section of the manuscript (pages 8-9, lines 295-307).

"AS and DM are co-founders of MedBiome, a clinical microbiomics company. The other authors have no competing interests to declare. "

RESPONSE: We have amended the cover letter as recommended. 

RESPONSE: There are ethical restrictions on publicly sharing the data from this study. During the informed consent process, participants were advised their data would only be shared with the approval of Dr Justin Presseau (Principal Researcher). Dr Presseau will ensure any reasonable request to access study data is granted – consistent with the process approved by the Children’s Hospital of Eastern Ontario Research Ethics Board. Data has been stored by the Ottawa Methods Centre at the Ottawa Hospital Research Institute (contactable at methodscentre@toh.ca). The methods centre can be contacted to obtain access to the data. 

RESPONSE: The Methods section of the manuscript has been updated, listing the Children’s Hospital of Eastern Ontario Research Ethics Board as the committee of record. Information on the verbal consent process has been moved, so that it is now listed immediately after listing the REB of record (pages 8-9, lines 295-307).

The reference list has been reviewed to ensure that it is complete and correct (pages 28-29).

Reviewers' comments:

Reviewer #1: Jen L. Olson, et al. investigated treatment fidelity in two related pilot clinical trials of an adjuvant treatment for pediatric-onset Inflammatory Bowel Disease. This is an important topic and very relevant in research and clinical practice. Authors have done an excellent job conceptualizing framework and the study team had access to a niche group of patient participants and have conducted interviews to understand barriers and enablers of performing trial-specific activities and of integrating those activities into daily life. The manuscript is well-written.

RESPONSE: We thank Reviewer 1 for their positive feedback. We agree this is an important topic that is highly relevant in research and clinical practice and feel it would make an important contribution to the existing literature. 

I have some suggestions to consider

- Consider revising introduction section to focus on treatment fidelity and process evaluations, and its impact on IBD. IBD pilot trials and participants information can be included in the methods section.

RESPONSE: The manuscript has been restructured as suggested. Specifically, descriptions of the pilot trials and participants have been relocated from the introduction to the methods section. Page 6 & 7, lines 231-258.

- It is not very clear how did the authors identify main themes and subthemes. Authors may want to elaborate on this in study methods.

RESPONSE: 

A description of the process undertaken to identify main themes and sub-themes has been added to the manuscript. This process involved reviewing the topics represented in each PPA and then grouping related topics into categories. Main themes represent overarching constructs including subthemes representing related but distinct barriers and enablers of performing trial-related activities (Page 10, lines 343-348).

- Did authors study other factors that could have influenced the study findings, like mental health diseases including anxiety, depression, etc, and other comorbid medical conditions in the participants and caregivers?

RESPONSE: Thankyou for raising the issue of mental health in youth with IBD. It certainly can be an issue. We have described the risk of mental illness among this population in the discussion section (page 26-27, lines 689-722). We did not include a targeted investigation of how mental illness and other conditions may have impacted on trial-related activities. In this study, our aim was focused on surfacing barriers and enablers of integrating trial activities into daily life. We recognise that for some participants, managing co-occurring conditions is a very important goal of daily life and although we did not include specific questions to surface barriers related to co-occurring conditions, the interview guide was broad enough to facilitate discussion of any factor considered salient by participants. Indeed, a few participants did very briefly discuss the impact of mental health on trial participation; however, this was not represented as strong concern (either in terms of the number of participants who raised it, or the degree of impact on trial activities for any participant). Given word count limitations, we focused on the themes and sub-themes with the strongest sentiment overall, and those described as having the biggest impact on performance of the trial activities. The potential impact of mental illness on the findings has been listed as an additional limitation in the Discussion section (Page 26, lines 689-722). 

- The discussion section needs a bit more work. Authors may want to include more evidence from other chronic conditions similar to IBD, and how patient experiences impact adherence to adjuvant therapy.

RESPONSE:

It is challenging to find particularly relevant evidence of how patient experiences impact adherence to adjuvant therapy in other chronic disease. For instance, barriers to adjuvant therapy for adult survivors of breast cancer doesn’t appear to be particularly relevant to the population under investigation. Nor does the literature around physical activity as an adjunct treatment in chronic disease. While we are not aware of a specific example of adjuvant therapy in chronic disease, we thought an analogy may be those children and adolescents with type 1 diabetes mellitus with requirement of prescription insulin along with adherence to dietary recommendations. The troubles faced in this population have been added to the manuscript (Page 22-23, lines 604-609). Additionally, we have made reference to the similarity of our findings with previous research investigating barriers to medication adherence among adolescents with IBD (page 24, lines 640-642).

Reviewer #2: Review Comments to the Author

Thank you for inviting me to review this manuscript reporting the qualitative evaluation of treatment fidelity alongside two related pilot trials of targeted therapies for IBD. The paper reports interesting and original findings that can help inform the interpretation of future clinical trials, with regard to trial outcomes and improving methods, and methodologies.

The authors have conducted appropriate qualitative methodology, including thematic analysis, and particularly relevant is the adoption of the multiple-goals perspective to interpret the themes. I have no major concerns on the manuscript, however the authors do need to provide greater explanation of the relation between trials, justification for sample size, and acknowledgement of study limitations.

RESPONSE: We thank the reviewer for their helpful suggestions. We have provided detailed responses to each point raised below and tracked any changes throughout the manuscript. This includes a more detailed description of the relation between trials (same aims, slightly different selection criteria with one trial recruiting newly diagnosed pediatric patients, and one recruiting those in the maintenance phase -page 6, lines 231-244), the sample size (rationale for wide range of recruitment target, given the anticipated heterogeneity of the sample, as well as the study design – pages 7- 8, lines 266-278), and acknowledgement of the limitations (data collection during COVID lockdowns which may have impacted the findings, the single time point design not allowing for assessment of changes in goals and priorities over time, and the potential for recall bias – pages 26-27, lines 674-722). 

My specific comments are as follows:

Title

Spell out acronym ‘IBD’ in the title.

RESPONSE: We have replaced the acronym in the manuscript title, now listed as “A qualitative evaluation of treatment fidelity alongside a pilot trial of a novel therapy for pediatric Inflammatory Bowel Disease” (Page 1 lines 1-3, Page 2 lines 35-38)

Abstract

There are scant details on the data collection methods. To the unversed, it is also unclear as to what ‘personal projects units of analysis’ refers to. You may wish to remove the ‘units of’, to provide standard PPA. Or succinctly explain/provide example.

RESPONSE: We have added additional information about the data collection methods to the methods section of the abstract (Page 2, lines 45-51). We have also removed reference to ‘units of’ PPA as suggested by the reviewer. 

Introduction

Page 4, lines 78-79: please explain briefly, why the prevalence of IDB is increasing worldwide.

RESPONSE: A brief explanation of the increased prevalence, attributable to compounding prevalence in Western nations and increasing incidence in newly ‘Westernized’ regions, has been included in the introduction (page 4, lines 88-91)

Very good rationale and explanation of the context of the qualitative (nested) research within, the larger project (i.e., two related trials). You may, however, wish to give the reader insight into the respective aims of the two trials. Although you introduce both trials on page 4 (lines 88-92), you only state what they both examined.

RESPONSE: We have edited the manuscript to reflect the shared aim of the trials more clearly, and to clearly indicate the trials only differed on inclusion criteria (RSP = newly diagnosed patients; MEND = clinically stable and in maintenance phase). Page 6, lines 230-244.

Method

Page 7, lines 141-143: greater justification is needed to the sample size. Between 15-30 interviews is a broad range; does this also include carers/parents, or just trial participants (e.g., children)?

RESPOSE: Justification of the broad range of interviews has been provided (mainly due to the anticipated heterogeneity of participants, in addition to the study design and planned analysis). We have also clarified that the specified range included caregivers, as well as trial participants. (pages 7 and 8, lines 266-278)

Lines 146-148: recruitment process – although you state when recruitment occurred, and that the research co-ordinator contacted potential participants, but there are no details as to how participants were selected, and from what database/recruitment pathway. These need describing.

RESPONSE: We have edited the manuscript to indicate that all participants from both pilot trials were invited to take part in the present study up until recruitment for interviews was ceased. (page 8, lines 282-291)

Also consider including author initials for specific tasks in the Method; you only do this occasionally and inconsistently. For example, the final sentence (page 9, lines 189-190) does not explain who resolved differences in interpretation.

RESPONSE: Author initials have been added throughout the methods section to indicate who was responsible for each task (Pages 6-9, lines 230-315)

Results

Effective use of quotes to support the generated themes. These align to the study aims and map out in the Discussion.

Discussion

Page 22, lines 471-472: ‘personally salient goals’; this is ambiguous. Please provide brief example(s) from your original findings

RESPONSE: Examples have been added, including doing homework to achieve good grades, and visiting with friends to have a fulfilling social life (Pages 24-25 lines 647-650)

The authors provide no limitations to this qualitative, nested research. Please provide on revision.

RESPONSE: We have revised the manuscript to draw attention to the limitations of this research. These include that the resea

---

## [Decision Letter · Decision Letter 1]

18 Jun 2024

A qualitative evaluation of treatment fidelity alongside a pilot trial of a novel therapy for pediatric Inflammatory Bowel Disease

PONE-D-23-30679R1

Dear Dr. Olson,

We’re pleased to inform you that your manuscript has been judged scientifically suitable for publication and will be formally accepted for publication once it meets all outstanding technical requirements.

Kind regards,

Yasin Sahin

Academic Editor

PLOS ONE

Additional Editor Comments (optional):

Thank you for the study. The authors did  an appropriate point-by-point response to  the reviewers. 

It can be accepted in its current form. I think that it will contribute to the literature

Reviewers' comments:

Reviewer's Responses to Questions

**Comments to the Author**

1. If the authors have adequately addressed your comments raised in a previous round of review and you feel that this manuscript is now acceptable for publication, you may indicate that here to bypass the “Comments to the Author” section, enter your conflict of interest statement in the “Confidential to Editor” section, and submit your "Accept" recommendation.

Reviewer #1: (No Response)

Reviewer #2: All comments have been addressed

2. Is the manuscript technically sound, and do the data support the conclusions?

Reviewer #1: Yes

Reviewer #2: Yes

3. Has the statistical analysis been performed appropriately and rigorously? 

Reviewer #1: Yes

Reviewer #2: Yes

4. Have the authors made all data underlying the findings in their manuscript fully available?

Reviewer #1: Yes

Reviewer #2: Yes

5. Is the manuscript presented in an intelligible fashion and written in standard English?

Reviewer #1: Yes

Reviewer #2: Yes

6. Review Comments to the Author

Reviewer #1: The authors have adequately addressed all my comments and questions. The manuscript is ready for publication in my opinion.

Reviewer #2: I feel the authors have addressed the comments of the two reviewers and in the revised manuscript the quality is of publishable standard.

7. PLOS authors have the option to publish the peer review history of their article (what does this mean?). If published, this will include your full peer review and any attached files.

Reviewer #1: No

Reviewer #2: **Yes: **James P. Gavin

---

## [Editor Report · Acceptance letter]

24 Jun 2024

PONE-D-23-30679R1 

PLOS ONE

Dear Dr. Olson, 

I'm pleased to inform you that your manuscript has been deemed suitable for publication in PLOS ONE. Congratulations! Your manuscript is now being handed over to our production team.

Kind regards, 

on behalf of

Dr. Yasin Sahin 

Academic Editor

PLOS ONE